

# Breeding and multiple waves of primary molt in common ground doves of coastal Sinaloa

Sievert Rohwer[1] and Vanya G. Rohwer[2]

[1] Burke Museum of Natural History and Culture and Department of Biology, University of Washington, Seattle, WA, United States of America

[2] Museum of Vertebrates, Cornell University, Ithaca, NY, United States of America

## ABSTRACT

For adult Common Ground Doves from Sinaloa we demonstrate that the primaries are a single molt series, which sometimes feature two (in one case three) waves of feather replacement. Such stepwise primary replacement is found in many large birds but, at 40 g, this dove is much the smallest species reported to have multiple waves of replacement proceeding through its primaries simultaneously. Pre-breeding juvenile Common Ground Doves never feature two waves of primary replacement. Juveniles usually have more than two adjacent feathers growing simultaneously and replace their primaries in about 100 days. In contrast adults, which extensively overlap molt and breeding, usually grow just a single primary at a time, and require at least 145 days to replace their primaries. Molt arrests are thought to drive the generation of new waves of primary replacement in a diversity of large birds. For adult Common Ground Doves, we found molt arrests to be strongly associated with active crop glands, suggesting that the demands of parental care cause arrests in primary replacement in this dove. For those adults with two primary molt waves, initiation of an inner wave was most frequently observed once the outer wave had reached P10. Thus, unlike reports for large birds, Common Ground Doves usually suppress the initiation of a new wave of molt starting at P1 when the preceding wave arrests before reaching the distal primaries. This assures that relatively fresh inner primaries are not replaced redundantly, overcoming a serious flaw in stepwise molting in large birds (*Rohwer, 1999*).

## INTRODUCTION

In large birds that fly while molting, the primaries are often replaced in multiple waves that proceed simultaneously through these longest feathers of the wing. Two mechanisms can generate these waves. In several orders of large birds, the primaries constitute a single molt series, but multiple waves of primary replacement are generated by molt arrests. Following arrests, molt is re-initiated at the next feather to be replaced in molt waves that have not reached the distal-most primary and also at P1, the first feather of the primary molt series in most birds. Thus, depending on the history of arrests and molt progression, large species that fly while molting may have from one to four waves of

Corresponding author
Sievert Rohwer, rohwer@uw.edu

replacement proceeding simultaneously through the primaries (*Rohwer, 1999*). This is known as *stafflemauser*, or stepwise molt (*Stresemann & Stresemann, 1966*), and has been documented in pelicaniforms (*Dorward, 1962*; *Rasmussen, 1988*), large herons (*Shugart & Rohwer, 1996*), accipitriforms (*Edelstam, 1984*; *Pyle, 2005*), and New World vultures (*Snyder, Johnson & Clendenen, 1987*), among other groups. Multiple waves of primary replacement are also achieved by dividing the primaries into two or more molt series. Multiple replacement series differ from stepwise molt in that they start and, with complete replacement, also stop, at predictable feathers within the wing. For large birds multiple molt series in the primaries has been well documented in albatrosses (*Langston & Rohwer, 1995*; *Rohwer, Viggiano & Marzluff, 2011*) and in Falconiforms and Psittaciformes (*Pyle, 2013*); multiple molt series are also found in some kingfishers (*Douthwaite, 1971*; *Hamner, 1980*), cuckoos (*Rohwer & Broms, 2013*), and a bell bird (*Silveira & Marini, 2012*).

At least in large birds, multiple waves of feather replacement in the primaries seem to facilitate more frequent replacement of the primaries because primaries may be growing at several loci that are separated by fully grown feathers. Some adaptation to increase the number of feathers grown simultaneously without creating large molt gaps in the wing is intuitively sensible because the time required to replace the primaries increases exponentially with body size and many birds over 1 kg do not replace all their primaries annually (*Rohwer, Rohwer & Ortiz-Ramirez, 2009*). In small birds for which the rules of primary replacement have been established, the primaries usually constitute as a single replacement series that are replaced in a single wave in a single bout of molting, without arrests. However, at least some small doves (*Pyle et al., 2016*; S Rohwer & V Rohwer, pers. obs., 2003) and the Mustached Tree Swift (*Hemiprocne mystacea*) exhibit stepwise primary replacement (*Rohwer & Wang, 2010*). Stepwise primary replacement is surprising in these species because they seem small enough to be able to complete primary replacement in a single bout of molting (*Rohwer et al., 2009*). What seems special about them is their extensive overlap in molt and breeding (*Rohwer & Wang, 2010*).

While arrests and the re-initiation of molt are documented to drive the generation of multiple waves of primary replacement in large birds (*Rohwer, 1999*), they are only suggested for small species. In Mustached Tree Swifts of the New Guinea region, which breed and molt year-round, primary replacement is stepwise, and the generation of multiple molt waves have been suggested to be associated with arrests caused by the demands of feeding young; however, evidence for this interpretation is lacking (*Rohwer & Wang, 2010*). Nonetheless, arrests associated with demanding periods of parental care could prove to be a general mechanism for generating multiple waves of primary replacement in small birds that show extensive overlap in molt and breeding. In contrast, reorganizing the primaries into multiple molt series would seem to be a more improbable evolutionary transition than the development of stepwise molting, so multiple series in the primaries may not characterize many groups of small birds.

We recently observed multiple waves of primary replacement in Common Ground Doves (*Columbina passerina*) from northwest Mexico. Here we quantitatively summarize their primary replacement to address two questions: (1) Do Common Ground Doves have stepwise primary molt or have they broken the primaries into multiple series? and (2) Is

breeding activity associated with arrests in primary molt, thus making arrests a possible mechanism for generating multiple waves of feather replacement in this small dove?

Distinguishing between stepwise primary replacement and having the primaries divided into multiple molt series can be non-intuitive from "snap-shot" data gathered from many individuals across the molting period (*Rohwer, 2008*). Yet the distinction is essential for understanding evolutionary transitions in primary replacement strategies across birds. The key is to understand that stepwise replacement is characterized by incomplete primary molts of unpredictable extent, with the result that, in the next episode of molting where molt reinitiates in the primaries distal to P1 varies across individuals. In contrast, the evolution of multiple molt series in the primaries seems designed to generate complete primary replacement by dividing the primaries into two or more short blocks of feathers, each of which is fully replaced in the same short period of time during a single bout of molting (*Rohwer & Broms, 2013*).

If Common Ground Doves have stepwise primary replacement, then three predictions should refute the hypothesis that the primaries are organized into multiple molt series. First, in stepwise molters, primary replacement should often be incomplete in a single bout of molting but rarely or never so in species with multiple molt series in the primaries. Second, in stepwise molters, juveniles undergoing their first bout of primary molt should possess only a single wave of feather replacement beginning at P1 while in species with multiple molt series in the primaries, juveniles molting for the first time should, like adults, initiate primary replacement at two or more fixed loci. Third, in adults of species that are stepwise molters, P1 will dominate as a site of molt initiation, but other sites of initiation will be scattered throughout the primaries, marking places where molt was reinitiated following an arrest. In contrast, when the primaries are organized into multiple series, molt will be initiated at a few consistent sites in the primaries, all with about equal frequency in composite summaries of "snap-shot" data (*Rohwer & Broms, 2013*).

We use data from 180 common ground doves from coastal Sinaloa, Mexico to describe their primary replacement rules. We document strong differences in the intensity of molt between juveniles and adults, show that molt can be stepwise in breeding adults, and provide the first evidence that arrests associated with parental care may be responsible for the generation of multiple waves of primary replacement in a small bird. Common Ground Doves, like other Columbiformes, regurgitate "crop milk", a liquid food consisting of protein, water and fat to their young, and the production of crop milk allowed us to assess how feeding young is associated with molt in breeding adults. Our results clearly show that this small dove overlaps molt and breeding more or less completely, can replace its primaries completely during the breeding season, and has evolved a mechanism to suppress the initiation of new waves of primary replacement at P1 until the preceding wave has reached the distal primaries. This discovery resolves an unexplained paradox in the Mustached Tree Swift (*Rohwer & Wang, 2010*), which, very likely, also has a mechanism for suppressing the immediate generation of new molt waves at P1 when the inner primaries are little worn.

## METHODS

Common Ground Doves are largely resident birds, with a range extending from the southern United States through Central America, across northern South America, and throughout the West Indies (*Bowman, 2002*). They are among the smallest doves, with a mass of about 40 grams (UWBM specimens from Sinaloa). Breeding in juveniles can occur when they are about six months old (*Johnston, 1962*; *Passmore, 1984*); we also confirmed breeding by three birds in partial juvenile plumage that were likely less than six months old. They do not defend territories and exhibit little aggression (*Nicholson, 1937*). Because the bursa of Fabricus is resorbed under the influence of sex steroids (*Mase & Oishi, 1991*), and because sex steroids are associated with aggressive behavior, the very limited aggression reported for Common Ground Doves may explain why they are slow to resorb the bursa and, thus, why both males and females in breeding condition often retain glandular bursas.

We studied Common Ground Doves on various communes in the Rio Fuerte flood plain between Los Mochis on the coast and El Fuerte near the foothills of the Sierra Madre Occidental in Sinaloa, Mexico. Prior to the development of massive irrigation, this region of coastal northwest Mexico was largely deciduous thorn scrub that leafed out in late summer with the arrival of the monsoon rains, but lost its leaves in November after the monsoons had ceased by the end of September. Even with massive conversion to irrigated agriculture (*Rohwer, Grason & Navarro-Siguenza, 2015*), ground doves nesting here are exposed to intense solar radiation while foraging and often while on their nests; thus, their feathers are subject to rapid fading from sunlight, as well as wear from the hard and thorny vegetation they use for nesting. This rapid fading and wear of new feathers enabled us to recognize blocks of newer and older feathers that were undoubtedly less than a year's difference in age.

Molt scoring and the analysis of the rules of primary replacement followed *Rohwer (2008)*. Growing primaries were given fractional scores representing, to the nearest tenth, their length as fully-grown feathers. New feathers were scored as 1 and old feathers as 0, but it is important to note that these age assignments are unlikely to be year classes in this data set; instead they represent activation classes of molt separated by unknown time intervals that likely were on the order of a few months. Where blocks of adjacent feathers represented different replacement classes, they were distinguished as newer or older than adjacent blocks. Rarely, adjacent groups of primaries were extremely worn and faded and assigned a score of ragged; these were likely feathers approaching a year of use that had been retained through the non-breeding season. The total molt score for a bird was the sum of the values for its growing and new or newer primaries. When molt scores from many individuals are summarized in tables, nodal feathers mark loci in the primaries where molt is initiated, either with the commencement of the annual molt or following arrests. In species with stepwise primary replacement and in species with the primaries organized into multiple molt series, stable nodes predominate in summary tables based on snap-shots of the molt status of many individuals; however, in stepwise molters, transient nodes, which mark sites where molt was reinitiated following an arrest, will be recorded at lower frequencies at unpredictable sites distal to P1 where molt restarted after an arrest.

We assessed breeding activity of doves that were captured and released by the presences or absence of crop-milk; for doves that were prepared as specimens, we noted the condition of the glands inside the crop, where vascularized crops with thickened walls indicated active milk production. We also recorded dates for a number of nests with eggs or young.

We have restricted all of our analyses to observations we made in coastal Sinaloa from 2005 through 2011, during the months of May through September. Most of our field work in Sinaloa was designed to study molt migrants (e.g., *Rohwer, Rohwer & Ortiz-Ramirez, 2009*; *Rohwer, Navarro & Voelker, 2007*; *Rohwer, Grason & Navarro-Siguenza, 2015*; *Rohwer, 2013*); consequently, we have scored molt for only 15 birds for the months of May and June, but many for the months of July through September. Some doves were collected and many more were scored for primary replacement before being released when work on other projects was slow. Time constraints, and not the condition of the bird, dictated whether birds were either collected or scored for molt and released; thus, our data should be an unbiased representation of the birds that we caught, even though our samples were obviously not collected systematically throughout the season. The number of birds scored for molt offers a good index of the relative amount of time we spent in the field in these months.

The approximately 40 ground doves that were collected were particularly important because they had associated extended wings that were used to measure both primary length and primary growth rates. Growth bands could not be seen in juvenile primaries; further, they were faint and difficult to see in most primaries of most adults and could not have been measured on specimens with folded wings. As part of our other studies of molt-migrants, we also collected or scored for molt a smaller sample of Common Ground Doves in Arizona; however, these birds were excluded from this study so possible geographic variation in the phenology of molting and breeding of this resident bird would not confound our results. Our Arizona observations were important, however, in demonstrating breeding by juveniles and multiple waves of primary replacement in adults.

Field work for this project was approved by the University of Washington Institutional Animal Care and Use Committee (protocol 4309-01). Collecting for this project was conducted under scientific collection permits issued by the Secretaría de Medio Ambiente y Recursos Naturales (SEMARNAT) to the Facultad de Ciencias (B Hernández-Baños FAUT 0169) of UNAM.

# RESULTS

## Primary replacement rules for juveniles

Most of the 44 juveniles that we captured and scored for molt were replacing primaries, suggesting that they likely begin this molt shortly after fledging (Table 1). Of the 44 juveniles we scored, just one had not yet commenced flight feather replacement and retained all juvenile primaries and secondaries, and one other had either arrested molt at P4, or was replacing primaries so slowly that active molt was scored as an arrest. Four juveniles were omitted from our count of 42 that were molting because they had two waves of molt; at least two of these were precocious breeders and the other two were uninterpretable. We

**Table 1  Summary of primary molt in juvenile Common Ground Doves by month.**

| Month | # juvs sampled | # replacing Ps | % replacing Ps |
| --- | --- | --- | --- |
| May | 1 | 1 | 100 |
| Jun | 5 | 5 | 100 |
| Jul | 10 | 9 | 90 |
| Aug | 7 | 6 | 86 |
| Sep | 21 | 21 | 100 |
| Totals | 44 | 42 | 95 |

**Table 2  Summary table for 40 juvenile Common Ground Doves that were growing primaries.** Note that the primaries are a single molt series with P1 nodal and P10 terminal and directionality always distal. Excluded are 4 juveniles with multiple waves of primary replacement, possibly associated with late fledging or precocious breeding.

| | P1 | P2 | P3 | P4 | P5 | P6 | P7 | P8 | P9 | P10 |
| --- | --- | --- | --- | --- | --- | --- | --- | --- | --- | --- |
| # nodal | 16 | 0 | 0 | 0 | 0 | 0 | 0 | 0 | 0 | 0 |
| # distal | | 17 | 19 | 13 | 13 | 8 | 13 | 11 | 7 | 4 |
| # proximal | | 0 | 0 | 0 | 0 | 0 | 0 | 0 | 0 | 0 |
| # terminal | 0 | 0 | 0 | 0 | 0 | 0 | 0 | 0 | 0 | 2 |
| # growing | 14 | 16 | 13 | 11 | 5 | 7 | 8 | 6 | 2 | 2 |

spent little time in the field in May and June, so the low numbers of juveniles scored in these months does not imply the production of few young during that time.

Without exception, juveniles replaced their primaries distally, starting at P1 (Table 2). P1 was strongly nodal and no other primary in our sample of 42 molting juveniles received a score of nodal, suggesting that arrests in the replacement of the juvenile primaries did not occur during the months of our field work. However, some late-fledged juveniles may be unable to complete their first primary molt, and these birds could be adding to the numbers of adults recorded as having multiple waves of primary replacement if we captured them in their second year. Our sample includes many more juveniles replacing inner than outer primaries, but good samples of juvenile ground doves growing primaries were available for all primaries except P8 and P9 (Table 2). Having samples for these distal most feathers requires recognizing remnants of juvenile feathers in other parts of the body, which we likely failed to do in some cases; this means that our sample of adults likely contains some older juveniles that were not recognized as such in the field. This problem suggests that the percentage of adults with multiple waves of primary replacement was underestimated.

## Primary replacement rules for adults

Primary replacement in adult Common Ground Doves is considerably more complex than it is in juveniles (Table 3). Of the 126 adults we scored for molt, 108 (86 percent) were replacing primaries; five of the 18 remaining adults had adjacent blocks of newer and more worn primaries in their wings, suggesting that they had arrested primary replacement. Just 17% of the adults that were growing primaries had two separate waves of molt. However, multiple waves of primary replacement are more common in adults than this figure suggests because some adults with arrested molt had two groups of newer primaries

**Table 3 Number of primaries growing for two classes of adults and for juveniles.**

| | Number of growing primaries | | | |
|---|---|---|---|---|
| | **1** | **2** | **3** | **4** |
| Adults, 1 wave | 102 | 15 | 1 | 0 |
| Adults, 2 waves | – | 14 | 4 | 0 |
| Juveniles, 1 wave | 12 | 16 | 6 | 4 |

**Table 4 Summary of primary molt in adult Common Ground Doves by month.**

| Month | # ads sampled | # replacing Ps | % replacing Ps | # replacing Ps in 2 waves | # with 2 active or inferred waves |
|---|---|---|---|---|---|
| May | 5 | 1 | 20 | 0 | 4 |
| Jun | 5 | 3 | 60 | 1 | 3 of 3[a] |
| Jul | 54 | 48 | 89 | 10 | 6 of 50[a] |
| Aug | 19 | 11 | 58 | 2 | 3 of 15[a] |
| Sep | 49 | 39 | 80 | 5 | 12 of 48[a] |
| Totals | 132 | 102 | 77 | 18 | 31 |

**Notes.**

[a] Primaries were not assigned ages in some birds, precluding inferring additional waves.

that were separated by one to several older primaries. These non-molting adults clearly had had two waves of primary replacement; when they are added to those with two active waves of molt, at least 21% of adults can be inferred to have been replacing their primaries in two waves.

Reviewers of this paper observed that the relatively limited numbers of adult Common Ground Doves with multiple molt waves suggests that adults showing two active molt waves could be late-fledged juveniles from the preceding year that were unable to complete replacement of their juvenile primaries. If this were the case, adults with multiple waves should be more frequent early than late in the breeding season, but there was no evidence of a seasonal decline in the frequency of adults showing two active waves of primary replacement ($X^2 = 2.37$; $p = 0.498$; Table 4, May and June poled).

As in juveniles, the direction of primary replacement is strongly distal in adults, with no cases of proximal replacement between any of the interior primary pairs (Table 5). However, molt arrests in adults complicate scoring of directionality. Thus, directionality scores between P1 and P2 were distal in six adults, but proximal in four. These four proximal scores are almost certainly scoring artifacts caused by P1 being too short to be compared for wear and fading with its adjacent inner primaries. Recall that feathers scored as new are relatively fresh, and thought to have been grown in that season and, further, that most cases of adjacent newer and older primaries probably do not represent year-class differences in feather age because molt arrests are common (see below). Thus, if P1 was growing, but too short to compare with P2 in its color and freshness, then, the inferred direction of replacement between P1 and P2 will be proximal when P2 is also scored as fresh. However, if we had been able to determine that P1 was newer than P2, directionality between this feather pair would have been scored as distal. All four of the anomalous directional scores
**Table 5  Summary table for 108 adult Common Ground Doves that were growing primaries.**

|            | P1  | P2  | P3  | P4  | P5  | P6  | P7  | P8  | P9  | P10 |
|------------|-----|-----|-----|-----|-----|-----|-----|-----|-----|-----|
| # nodal    | 12  | 0   | 0   | 0   | 0   | 2   | 3   | 0   | 2   | 1   |
| # distal   |     | 6   | 8   | 16  | 29  | 29  | 40  | 38  | 26  | 12  |
| # proximal |     | 4[a]| 0   | 0   | 0   | 0   | 0   | 0   | 0   | 6   |
| # terminal | 0   | 0   | 0   | 0   | 0   | 0   | 3   | 1   | 1   | 9   |
| # growing  | 8   | 4   | 8   | 15  | 19  | 21  | 25  | 19  | 12  | 11  |

**Notes.**

[a]These four exceptions to distal replacement are likely unavoidable scoring problems caused by P1 being a pin or short feather. P1 was presumable initiating a new wave in each case, but was too short to be compared with its neighboring inner primaries, which were little worn.

between P1 and P2 were cases where P1 was a pin feather or a short brush just emerging from its sheath; thus, none of these four cases legitimately imply proximal replacement between P2 and P1 because the age of P1 could not be assessed by feather color. Further, we saw no evidence of P1 and the outer series of secondaries being part of a single molt series (S Rohwer & V Rohwer, 2017, unpublished data).

A similar problem arises with arrests at P9. When molt is reinitiated at P10 and the feather(s) immediately proximal to it are scored as old, then, by definition, the inferred direction of replacement between P9 and P10 is proximal, readily accounting for the 6 proximal directionality scores between P9 and P10 as artifacts of arrests (Table 5). In contrast, the 14 distal directionality scores between P9 and P10 are unambiguous. Summarizing, all of the proximal directionality scores between primary pairs P1/P2 and P9/P10 can be interpreted as scoring artifacts of the frequent arrests that characterize primary replacement in adult Common Ground Doves; directionality between interior primary pairs was exclusively distal.

Unlike the situation in large birds, scattering of nodal and terminal feathers from P6 to P9 does not only reflect the generation of multiple waves, but also the locations of arrested molts (Table 5). Nodal feathers mark the beginning of waves of replacement in the outer primaries. However, they can only be recognized as nodal when an arrest has been long enough that the feathers replaced in the preceding bout of molt have accumulated noticeable wear or fading. Then, the primary at which a molt wave is re-activated will be newer than its proximal neighbor and thus be scored as nodal. That the recognition of nodes in the distal primaries requires arrests of sufficient duration to recognize older feathers surely means that our data underestimates the number of adults inferred to have arrested primary replacement. The same scoring problem arises when a proximal wave of replacement extends into older feathers from a previous arrest. In these cases the distal-most growing feather in the proximal wave of active molt will be scored as terminal because both of its neighboring feathers will be older than it is. To reiterate, the scattering of feathers scored as nodal or terminal in the distal primaries confirms, but likely under-represents the frequency of arrested waves of primary replacement in ground doves from Sinaloa.

Data linking breeding attempts with arrests in primary molt in adults is shown in Table 6 where we compare crop gland activity with primary replacement. Many adults that were replacing primaries had active crop glands, so molt is not always arrested when young are being fed (Table 6). Nonetheless, arrested primary replacement was strongly associated

**Table 6  Association between primary replacement and active crop glands in adults.**

|  | Molting Ps | Not molting Ps | % molting |
|---|---|---|---|
| Crop gland active | 8 | 6 | 57 |
| Crop gland not active | 30 | 1 | 97 |

with having an active crop gland (2-tailed Fisher's exact $p = 0.002$), suggesting that adults providing crop milk to young may arrest their molt to conserve energy. In large birds with stepwise primary replacement, these arrests are thought to generate multiple waves of primary molt. When primary replacement reinitiates, it picks up where it left off in the distal primaries and also starts anew at P1, thus generating two or more waves of primary replacement.

Common Ground Doves stand in striking contrast to this understanding of stepwise primary replacement in large birds. As we show in Fig. 1, most arrests do not result in the generation of a new wave of primary replacement at P1 when molt is resumed. Instead, new waves of molt are not initiated in the inner primaries until the prior replacement wave has reached P10, or sometimes P9 (Fig. 1).

To show this we have used three common patterns of primary scores. First are birds showing blocks of new feathers in the inner primaries (the outermost of which may be growing), and old feathers in the outer primaries, characteristic of a single wave of primary molt. For these cases, the outermost new or growing primary was plotted, giving the blue line in Fig. 1. This line shows that new waves are seldom initiated until molt has reached distal primaries P9 and P10. All lines in this figure start with P3 because primary replacement had to have progressed far enough into the wing for a block of new inner primaries to be discernable. In contrast, multiple waves are represented by blocks of new or growing feathers in the inner and outer primaries, separated by a block of old feathers in the middle primaries; in this case we plotted the outermost new or growing primary (Fig. 1, black line). This black line clearly shows that blocks of new inner primaries are rarely seen until molt in the distal block of new primaries has reached P9 (just three cases) or P10 (19 cases). Together, the contrasts between the blue and black lines strongly suggest that inner waves are suppressed, even if there are arrests, until primary replacement has reached the outermost primary. Finally, the orange line confirms this conclusion because it represents cases where molt could be seen to have been arrested in the inner primaries (a block of new feathers) and restarted (a block of *newer* middle primaries), while the outer primaries were scored as old. In this case, we plot the location of the arrests, which we scored as the break between the inner new and outer *newer* feathers, and not the outer most growing primary. This line is important because, while surely underestimating the frequency of arrests (because new and *newer* blocks of feathers were not always distinguished), it clearly illustrates primary replacement can arrests and restart from P5 to P7 without a new molt wave initiating at P1. The take-home message of Fig. 1 is that arrests in primary replacement do not initiate new waves of replacement at inner primaries unless molt has proceeded to the distal-most primaries.

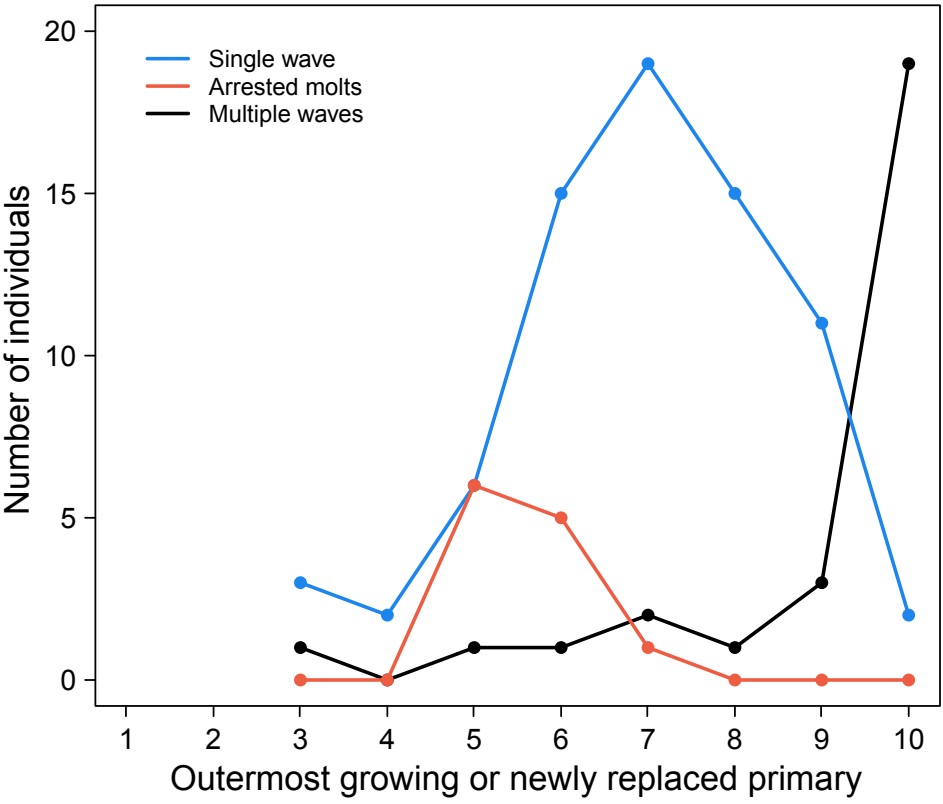

**Figure 1** **The progression of molt in outer molt waves corresponds to the initiation of inner molt waves in adult Common Ground Doves.** Individuals with a single molt wave (blue line) are commonly observed until molt has progressed to P9 and P10. By contrast, the frequency of individuals with two molt waves (black line) dramatically increases once the outer molt wave has reached P10. Individuals that arrested and re-initiated molt without starting a new series at P1 (orange line) show that arrests do not activate a molt series at P1, as occurs in large birds. All lines start at P3 because our scoring scheme required groups of feathers for determining feather ages. See text for more information.

## Molt breeding overlap and molt duration

Our records suggest that primary molt and breeding overlap extensively in Common Ground Doves in Sinaloa (Table 4). We found nests and birds with active crop glands from May through September and more than 50% of adults examined were molting in all of these months except May (Table 4). The long breeding season and the extensive overlap of molt and breeding in adult ground doves suggests that there should be little relation between day of year and primary molt score for adults. When all the data are plotted (Fig. 2), there is a significantly positive trend (slope = 8.06, $p < 0.0001$; $r^2 = 0.27$), with replacement of the primaries being closer to completion in September, which is likely near the end of the breeding season. This does suggest that molt is scheduled to replace all primaries during the breeding season. It is critical to note, however, that the distribution of points strongly suggests that the data is largely a composite of two lines that are almost flat, representing the months of July and September when many adults were scored for molt. This is indeed the theoretical expectation when some birds are completing molt

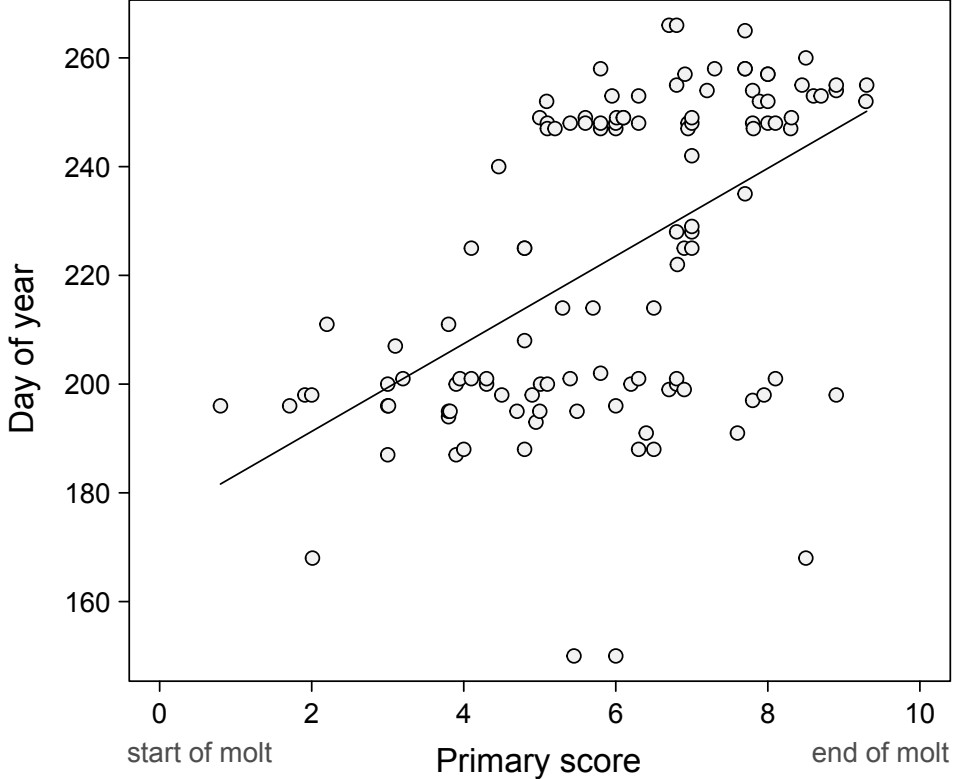

**Figure 2** **Progression of molt in adults.** *Pimm (1976)* regressions fail to correctly estimate the duration of primary replacement when start and finish dates overlap broadly in the studied population, as is the case for these doves. Thus the progression of molt in adults largely reflects birds in various stages of molt in the months of July and September when most of our birds were scored.

as others are just beginning (*Rohwer & Broms, 2012*; *Rohwer, 2013*). Thus, the presence of two "flat" regression lines in Fig. 2 is evidence that, as in other species, where molt is little synchronized among individuals in a population, neither regression nor maximum likelihood estimates of molt duration are reliable. Almost "flat" lines in Pimm regression plots suggest extremely rapid molts, something that ground doves do not do. Such "flat" regression lines likely characterize many species that show extensive overlap in molt and breeding, and they also occur when some individuals have completed primary replacement before others have started, as is true in Painted Buntings, *Passerina ciris* (*Rohwer, 2013*).

Following *Rohwer & Broms (2013)*, we estimated the time required to replace the 10 primaries in adult Common Ground Doves from the summed length of the primaries, primary growth rate, and the number of primaries growing simultaneously. Our measurements for primary lengths and growth rates are summarized in Table 7. For birds in active molt, the mean number of primaries growing was 1.62 for adults and 2.52 for juveniles (weighted averages computed from data in Table 3). Assuming no interruptions in the molt, which of course is not true in adults, but the rule for juveniles,

Table 7 **Mean primary lengths (standard deviations in parentheses) and growth rates for Common Ground Doves.** Primary growth rates were measured from growth bands in two or three feathers; standard deviations are given when there were three measurements.

| | Primary length | | Primary growth rate (mm/d) |
|---|---|---|---|
| | Females ($n = 8$) | Males ($n = 6$) | (Sexes combined) |
| P1 | 51.8 (0.75) | 54.1 (1.69) | 2.88 |
| P2 | 52.9 (1.10) | 54.8 (1.47) | 2.72 |
| P3 | 53.7 (0.76) | 55.5 (1.38) | 2.68 (0.63) |
| P4 | 55.1 (1.28) | 56.9 (0.67) | 2.90 (0.18) |
| P5 | 56.9 (1.02) | 59.1 (0.92) | 2.78 |
| P6 | 61.0 (1.29) | 63.0 (1.27) | 2.44 |
| P7 | 63.5 (0.58) | 64.7 (1.48) | 3.61 |
| P8 | 63.6 (0.89) | 65.2 (1.17) | 3.03 |
| P9 | 62.5 (0.84) | 63.8 (0.75) | 3.15 (0.38) |
| P10 | 59.4 (0.67) | 60.3 (0.82) | 3.01 (0.29) |

the number of days ($D$) required to replace all the primaries is

$$D = L/(G \times N),$$

where $L$ is the summed length of the 10 primaries, $G$ is primary growth rate measured in mm/d, and $N$ is the mean number of primaries growing simultaneously. Using this equation, adult males are estimated to require 149 days to replace their primaries, adult females 145 days, and juveniles 101 days. Because birds were breeding during the extent of our field work from May through September, we do not know how much time they have to spend in molt. Nonetheless, these figures suggest that adults could replace all their primaries in less than the five months when we know they are breeding, so most adults should be able to complete their molt while they breeding, even with arrests during intense periods of parental care. Juveniles take considerably less time to molt but, if foraging is difficult during the dry months of winter, late-fledged juveniles may not be able to replace all their primaries before their first breeding season.

## DISCUSSION

We show that the primaries of Common Ground Doves constitute a single molt series and are replaced distally from P1 to P10. In contrast to juveniles, adults often had two active waves of primary replacement and they usually had just one feather growing per wave. Molt arrests are thought to be the mechanism by which multiple waves of primary replacement develop in large birds (*Rohwer, 1999*). Although, we document that molt arrests in adult Common Ground Doves are strongly associated with active crop glands, indicative of adults feeding young, arrests in primary replacement seem not to result in a new wave of molt starting at P1 unless earlier bouts of primary replacement have reached P9 or P10. Juveniles replace their primaries in a single wave that starts at P1 and no juvenile had two waves of primary replacement, unless they were breeding precociously, as rarely occurs in this species (*Johnston, 1962*; *Passmore, 1984*). Stepwise primary replacement is common in

large birds, but this represents only the second well documented case of stepwise molting in a small bird, the other being the Mustached Tree Swift (*Rohwer & Wang, 2010*). At 40 grams, Common Ground Doves are the smallest species for which stepwise molting has been documented.

Stepwise primary replacement seems adaptive in large birds because it facilitates the generation of multiple waves of primary replacement in species with too little time to replace all their primaries annually (*Stresemann & Stresemann, 1966*; *Rohwer, Rohwer & Ortiz-Ramirez, 2009*) or between breeding attempts in species that breed year-round (*Ashmole, 1968*). The key feature of stepwise primary replacement is that molt recommences where it was arrested and also at P1 (*Rasmussen, 1988*; *Shugart & Rohwer, 1996*; *Rohwer, 1999*). Recommencing at P1 generates considerable within-species flexibility in primary replacement, both across individuals and populations. Where the time available to replace primaries is limited, the generation of multiple waves eventually results in more primaries being renewed in each bout of molting, while more time for primary molt should generate fewer waves of replacement, resulting in fewer molt gaps in the airfoil.

The discovery of stepwise primary replacement in small species that likely could replace all their primaries annually begs the question of whether stepwise primary replacement is adaptive in small species. We suspect it is because we see no reason that arrests must always be followed by the generation of a new molt wave starting at P1. Although this is a key feature of stepwise molting in large birds, reinitiating molt at P1 seems not to occur in Common Ground Doves until replacement of the primaries has progressed at least to P9 or P10. Further, at least in some shorebirds, arrests are frequent, but molt picks up where it was arrested, without starting over at P1 (*Pienkowski et al., 1976*; *Summers et al., 1989*; *Remisiewicz et al., 2009*). These shorebirds are also small species that usually can replace all their primaries during the nonbreeding season. The tendency for Common Ground Doves to suppress replacing P1 following arrests unless the preceding molt wave has progressed into the distal primaries further suggests adaptation to stepwise molting in this small bird. We note, however, that satisfactorily addressing the adaptive significance of stepwise molting in small species will surely require far more descriptions of molt in small to mid-sized species that are not closely related.

In large birds, stepwise molting often results in the inner primaries being replaced two or more times before the distal-most juvenile primaries are replaced for the first time (see diagram in *Rohwer, 1999*). This seems remarkably maladaptive because the outer primaries wear considerably more than inner primaries, yet are the last to be replaced in young birds. Prior to this study, the only mechanism identified for overcoming this problem was "omissive" molts, reported for shags and large herons (*Rasmussen, 1988*; *Shugart & Rohwer, 1996*). In omissive molts either P8, P9 or P10 is lost out of sequence in some (but not all) juveniles, thus setting up replacement of outer primaries before the first wave of molt replacing juvenile primaries has progressed into the distal primaries. Omissive molts have remained perplexing because only some individuals show such replacement, and because the primary that is replaced varies. These conundrums may be resolved by the recent discovery that some birds have the ability to preferentially replace broken primaries or rectrices (*Ellis, Rohwer & Rohwer, 2016*). If omissive molts are generated by the preferential

replacement of extremely worn primaries, then the fact that not all juveniles show this pattern of replacement in their outer primaries and that the replaced primary can be any of three different feathers is resolved: very worn outer primaries could be replaced out of their normal sequence; further, if the replaced feather is not the terminal primary, its replacement could also start a distal wave of primary molt.

Our results for Common Ground Doves have revealed another, previously unknown mechanism for avoiding unneeded frequent replacement of the inner primaries in birds with stepwise primary replacement. Somehow, ground doves largely suppress initiation of new replacement waves at P1 until the distal wave has proceeded to P9 or P10. Because inner primaries wear more slowly than distal primaries, this suppression of new waves starting at P1 largely eliminates frequent replacement of the inner primaries, something that has not, to our knowledge, been reported for large birds (*Rohwer, 1999*). This difference in when and how inner replacement waves are generated in Common Ground Doves compared to large birds, suggests that this stepwise-like molt may be better thought of as "stepwise with suppression". Clearly, some other mechanism regulating the activation of molt waves frees doves from initiating inner replacement waves with the re-initiation of arrested outer waves. Common Ground Doves were serendipitously well suited for this discovery because the number of primaries between the distal growing feather in birds with two molt waves is little affected by differences between feather lengths and growth rates, which vary little across their primaries (Table 7). Thus, data from individuals with two waves of molt could be used for this test, without differences in the time it takes to grow different feathers strongly biasing results.

Delaying the development of new waves of feather replacement starting at P1 also explains the surprisingly greater frequency of growing feathers in the outer than in the inner primaries (Table 4), which, initially, seemed anomalous to us. If new waves that follow too closely on the heels of a previous wave are suppressed, then molt summary tables (which aggregate molt scores across many individuals) should show more growing primaries in the distal primaries. This is partly because outer primaries are longer and take more time to grow, but that likely has little effect in Common Ground Doves because the longest primaries (P7 or P8) are only 12 mm longer than P1 (Table 7). Instead, this higher frequency of growing feathers in the distal primaries probably results from new waves of replacement being suppressed in the inner primaries following arrests until molt in the outer primaries has reached at least P9 or P10.

This pattern of more active molt in the distal primaries was also discovered in Mustached Tree Swifts, but was perplexing to *Rohwer & Wang (2010)*. Now it seems likely that the suppression of new waves starting at P1 until the next distal wave has progressed into the distal primaries could be a general solution to the problem of excess replacement of inner primaries in small birds with stepwise-like primary replacement. In Mustached Tree Swifts the outer primaries are progressively much longer than the inner primaries, but the strong break in the frequency of replacement occurred between P5 and P6, with P6–10 showing about twice as many growing feathers as P1–5. Because this was a more or less dichotomous break in frequencies, despite progressive length differences in the primaries, we suggest

that Mustached Tree Swifts may also suppress the commencement of new replacement waves at P1 until molt in the outer wave has progressed at least to P6. Another species with data consistent with "stepwise with suppression" molts are Wood Pigeons (*Columba palumbus*), where late-hatched first year birds, as well as adults that had two waves of replacement in their primaries, had 5–6 newly replaced feathers between waves (*Boddy, 1981*), suggesting suppression of an inner wave until the outer wave has reached the distal primaries. Studies of marked and recaptured birds with known breeding histories will be essential to support our suggestions that arrests are responsible for generating multiple waves of primary replacement, and that new waves are suppressed when an arrest occurs before the initial molt wave has progressed to P6 or beyond.

More broadly, pigeons and doves exhibit remarkable flexibility in their molts. Mourning Doves (*Zenaida macroura*) and White-wing Doves (*Zenaida asiatica*) replace their primaries distally from P1 to P10, in a single series with apparently few arrests (*Otis et al., 2008*). By contrast, we have seen multiple waves of replacement in small *Ptilinopus* doves collected in the Solomon Islands, and *Pyle et al. (2016)* report two waves of feather replacement in two *Ptilinopus poryphyraceus* from American Samoa, suggestive of stepwise molting. Stepwise molts, presumably generated by arrests for breeding, are also reported from some Band-tailed Pigeons, *Patagioenas fasciata* (*Silovsky et al., 1968*) and Wood Pigeons (*Boddy, 1981*). These observations suggest flexibility in the scheduling of annual molts relative to breeding opportunities. When breeding conditions are favorable, some individuals in active molt arrest and then resume their molts at multiple feather loci, while other individuals simply continue their annual molts without arrests. Stopping and starting molt likely increases the potential for creating asymmetries between wings, but any costs of asymmetric molts appear outweighed by the benefits of additional breeding attempts. Pigeons and doves, with their specialized crop glands for feeding young, may be especially prone to arrested molts when the food demands of young are high.

Many of the regions in coastal Sinaloa where we worked have been converted to irrigated agriculture (*Rohwer, Grason & Navarro-Siguenza, 2015*), which provides extensive habitat and more predictable mesic areas around fields that are excellent for breeding by Common Ground Doves. This also means that breeding opportunities for ground doves may extend for longer periods of time and be less confined to the annual late-summer monsoon (*Comrie & Glenn, 1998*) than would have been the case prior to the 1970s, when reservoirs and extensive systems of cement-lined canals began to be developed in coastal west Mexico (*Rohwer, Grason & Navarro-Siguenza, 2015*). The possibility that landscape-level changes has affected the phenology of molt and breeding or the extent of molt-breeding overlap could be assessed by comparing ground doves in parts of Sonora where monsoon rains continue to drive the annual vegetation cycle in the absence of irrigation.

As more tropical species are examined that exhibit extensive molt breeding overlap, many more small species will likely be discovered to replace their primaries in multiple waves, either in a stepwise fashion, or by dividing the primaries into multiple series. Distinguishing between these two strategies requires data on substantial numbers of actively molting birds

summarized in tables similar to those presented here. As more such studies accumulate, the comparative studies they will support should help illuminate both the evolutionary history and lability of primary replacement patterns across birds and how primary replacement strategies are integrated into other major life-history features of birds.

### Funding

Support for fieldwork in northwestern Mexico came from the Burke Museum Endowment for Ornithology, and grants from Hugh S. Ferguson, the Nuttall Ornithological Club, and region 6 of the USFWS. The funders had no role in study design, data collection and analysis, decision to publish, or preparation of the manuscript.

### Grant Disclosures

The following grant information was disclosed by the authors:
Hugh S. Ferguson.
Nuttall Ornithological Club.
USFWS.

### Competing Interests

The authors declare there are no competing interests.

### Author Contributions

- Sievert Rohwer and Vanya G. Rohwer conceived and designed the experiments, performed the experiments, analyzed the data, contributed reagents/materials/analysis tools, wrote the paper, prepared figures and/or tables, reviewed drafts of the paper.

### Animal Ethics

The following information was supplied relating to ethical approvals (i.e., approving body and any reference numbers):

Collecting for this project was conducted under scientific collection permits issued by the Secretaria de Medio Ambiente y Recursos Naturales (SEMARNAT) to the Facultad de Ciencias (B Hernandez-Banos FAUT 0169) of UNAM.

### Data Availability

The raw data is provided as Data S1.

### Supplemental Information

Supplemental information for this article can be found online at http://dx.doi.org/10.7717/peerj.4243#supplemental-information.

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
