# Peer review of "Breeding and multiple waves of primary molt in common ground doves of coastal Sinaloa"

_PeerJ, doi:10.7717/peerj.4243_

## Round 0.1 · original submission · Minor Revisions

I have received two detailed reviews and feel your manuscript is suitable for publication after minor revisions.

My own concern is very minor but related to comments about your primary data (especially reviewer #2's comment about including better descriptions of the information). I like to include a "README" worksheet in spreadsheets that includes a text box with a detailed description of the information. That approach does not take much work and it allows readers to move back and forth between data and description of data simply by clicking the tab.

I'd really like to see you try something like that. A separate README would also work, but I find that separate files can cause confusion. Beyond that, I feel the reviewer's comments were both valuable and clear and I hope you will revise the manuscript by addressing them.

·

Basic reporting

The paper is fine and relatively clearly written in English but would benefit from a widening of focus to explore a wider view of the scope of the difficulties in scoring molt in a field study were there will always be unknowns related to seasonality or lack of, lenght of breeding/non-breeding period, molting/non-molting periods, multiple broods related to all of the former. Two additional references not included I noted in general comments.

Both authors have extremely strong backgrounds and are leaders in this field of research

The paper is structure professionally

The paper is self contained for the narrow look at stepwise molt given the limitations induced by unknown is this type of field work.

Experimental design

Methods could use some clarification and checking of numbers of cited in text based on supplemetal data tables, see General Comments.

Validity of the findings

Data are intriguing and the paper provides unique data that will become a much cited reference for future research. I spent several hours checking and proposing what if scenarios and I'm sure others will do the same. With this in mind, the authors should consider a more open framework to outline alternatives rather than forcing data into Stafflemauser and arrested molt for breeding scenario. Consider patterns from arrests of molt from seasonality, for breeding, combinations of the two, age structure of populations, and arrests with or without re-initiation because feathers have a latency to molt. And perhaps most interesting, why juveniles are replacing primaries so soon after growing them. But I'd like to stress that this is my opinion and should be construed as necessary.

Additional comments

First task is to proof and spell check supplemental data table and verify numbers. For example, the first sentence of Results indicate there were 50 juveniles when there are 44 in the supplemental table. Methods indicate 175 birds when there are 182 in supplemental table. See comments. These are minor inconsistencies perhaps reflecting birds included or excluded for various reasons but clarification would be helpful. One colorful addition is the conditional formatting option in Excel that color codes molt values - I left this in on the Supplemental Table sent to the Associate Editor (I couldn't attach the Excel sheet as a pdf). When completed, move Table 6 to Table 1 adding ages to provide an introductory summary of data.

Data in the manuscript are crucial to further understanding patterns of molt. However, data were subjected to narrow interpretation and need a more careful and thorough analysis and summaries without preconceived notions regarding Stafflemauser. There seems to be sufficient info in Stresemanns’s, Pyle, BNA, especially Rivera-Milan, F. F. 1996. (Nest density and success of columbids in Puerto Rico. Condor no. 98:100-113) to assume molt is slow sequential descendant with arrests possible, but it is useful to verify this as the ms does. Ground-dove data seem similar to woodpigeons (cited in ms) except for unknown impacts of the length of seasons on ground-doves. A relevant paper to include is Ashmole's (1968) study of Fairy Terns stepwise molt, near year-around breeding, and arrests for breeding.

There may be an inconsistency in consideration of Stafflemauser. The discussion of omissive molt and the multiple replacement of proximal primaries (or Ps) once or twice while setting up Stafflemauser are said to be maladaptive in the paper. The last sentence of abstract suggests the latter is the “single greatest flaw”. If so, then is Stafflemauaser simply an artifact of incomplete P molt? Alternatively, if Stafflemuaser is an adaptation, an incomplete P molt is a prerequisite to setting up waves. Thus the retention of outer Ps for a year or two are putative costs that immatures incur in order to benefit from Stafflemauser as adults. As for multiple replacement of inner Ps, I believe this refers to yearly replacement which birds would do typically so not seeing how it is a flaw or cost. Once waves are set up, benefits are speculative depending on how many Ps are replaced as adults. With complete replacement, benefit might be more efficient replacement with 3 waves than in a single wave of molt. With incomplete replacement, the outer Ps can be replace every other year. The discussion of omissive molt and multiple replacement of proximal Ps need some minor rewriting to reflect this overview.

However in revising consider that if data are filtered to only show the 13 adults that are molting P10, the interpretation of Stafflemauser is suspect. A prediction from Stafflemauser is that waves are synchronized more or less. If so, if P10 was molting and appears to be a nodal feather (proximal neighbor was old (0=old), then proximal and medial waves would also be re-initiating with a similar number of Ps renewed. However only two wings, UWBM 82497 and CWS 9167 appear as predicted with similar numbers of new (=1) Ps in proximal and distal waves suggesting a lack of synchrony on re-initiation. In others the proximal wave has progressed many more Ps than would be expected by an assumed roughly synchronized re-initiation of waves. Perhaps this results from proximal waves abutting more distal leaving a continuous sequence of similar wear class Ps as discussed in the ms? Alternative, P10 is molted asynchronously from proximal Ps or as Broody for woodpigeons.

P1 appears to be similar. Of 9 growing P1, 5 appear to initiate at P2 so it appears P1 molted independently or perhaps with secondaries?

So for ground-doves, is the molt observed a result of a slow (150 days in ms, 200 days in Rivera-Milan 1996) simple sequential descendant pattern that takes place yearly with some arrest & re-initiation or an adaptation/modification worthy of being considered Stafflemauser similar to Fairy Terns (Ashmole 1968). Data indicated that only 30% of ground-doves have multiple waves suggesting the pattern might be an artifact. This might be a better way to frame hypothesis testing rather than the idea that there are distinct series in primaries (similar to secondaries). Add that since 70% of adult have a single wave, multiple waves may be SY and TY (second or third year) birds ironing out the wrinkles to set up a single slow descendant wave as adults.

For clarity, redo the direction summary using pairs of growing primaries for P2-9, excluding P1 & P10 as focal feathers because they lack prox and distal neighbor Ps, respectively. This will affirm (or reject) that molt is directional. Then summarize waves more simply than Table 3 & 4. The hypothesis to disproof appears to be that are set nodes, e.g. P1, P4, P8 and therefore node and terminus of waves would be predictable. The difficulty here is that this is only true if all Ps are replaced in which case the Stafflemauser and the hypothetical set-node Stafflemauser would be indistinguishable in adults. With incomplete replacement, assuming re-initiation, there would be no predictability of either accept perhaps a node at P1. Most the set-node patterns in primaries start in middle or proximal Ps and go both ways. Owls, not mentioned, likely have the record for slow molt. Regardless of direction, terminal and nodal Ps are predictable only if all Ps were replaced. Incomplete molt results in unpredictability.

On constraints from breeding and crop milk for feeding young. Milk is fed to young nestlings but then I believe adults shift to regurgitated seeds as young grow so not sure that an active crop signals an energetic constraint from feeding young and manifest as a molt arrest. If it takes 150-200 days for adults to molt all Ps, and there are multiple broods, seems more likely the extremely slow molt is an adaptation to prolonged window when breeding & molt is possible. And assuming molt is extremely slow, there is likely a latency in some wings to completion of one P and dropping of the (distal?) neighbor, which might appear to be an arrested molt (a check for molting secondaries might provide a clue here) (also mentioned by Boody for woodpigeons). One could turn the discussion around and suggest that so many were molting (~80%) that they molt continually while breeding and most successful adults do both without arrest. Leaving multiple waves an artifact of incomplete molt or a rudimentary Stafflemauser. A latency to molt hypothesis might be presented as a contrast to Stafflemauser in highly seasonal species. Once renewed a feather will not be physiologically ready to molt until it is X months old thus stopping or merging waves and introducing the idea for what was observable in P1s and P10s.

An interesting offshoot is why are juveniles molting primaries so early? If the poor feather quality, then the juvenal feathers should be able to be identified and if no obvious different wear categories, then Ps probably aren't juvenal. E.g., 5 "juveniles" had class 0 (old) primaries in the wing distal to J primaries. These would have to be scored as "J" or "J0" if the birds were juveniles. Perhaps they SYs (second year) that have gone through one non-molting period. A complication, briefly mentioned is that if a pair had multiple broods over a breeding window, the first fledged would likely complete molt while there would be a progressively fewer completing molt in subsequent broods with the last fledged only molt few Ps. Following a non-molting period, if it exists, the SY birds would likely be classed as adults unless juvenal plumage hints were retained. This would create a scoring nightmare with birds in all stages of complete, incomplete, and some erratic molt patterns with the underlying pattern of sequential descendant molt. To further non-temperate molt studies introduce molt latency as a controlling factor and provide a simple figure showing what P wear categories might look like with staggering introduction of juveniles that undergo a period of arrest. Depending on the age structure of population of ground-doves and similar species, SY birds might be most common. Also suggest as Ashmole did that researchers might punch primaries if they had an opportunity to recapture the birds.

Figure 1. Number of feathers between the outermost growing primaries in the 15 adults that had two active waves of primary replacement. (in supplemental table there are 19 adults with 2 active waves, 17 scored and 17 shown in figure)

Figure 2 legend. "sailed population" an autocorrect for "Sinaloa population"?

Table 1 Totals 43 41 95% should be 44 total sampled and 42 replacing primaries

Table 2 check counts of growing, correct except P1 looks like it is 14 rather then 13

Table 3 delete simultaneously as it isn’t possible for a single feather

Table 4. Summary table for 100 adult Common Ground Doves that were growing primaries. But there are 112 in supplemental table, some were not included? Previously noted UWBM 90871 incorrectly designated no molt but P1 is growing so make P1 growing 9 rather than 8, check others growing, most are correct but there may other discrepancies depending on how many are included

Table 6 132 adults in table 138 in supplemental data, 28 not molting (one –UWBM 90871 - molting at P1 designated not molting in the supplemental table)

Throughout change Ground Doves or Ground-Doves

Some images of wings showing various wear classes of primaries would be useful for future reference

A few comments on the Supplemental data table and second worksheet summarizing growing P1 and P10 were sent to the Associate Editor because formatting was lost while saving as a pdf. I can forward these to the authors on request.

Reviewer 2 ·

Basic reporting

The manuscript is written clearly and overall well referenced, and the introduction provides good background on the subject. The structure conforms the journal’s standard. Raw data were provided but need more descriptions of symbols they used (for example, I was not sure what “r” means in the primary molt score columns). Also, I felt that some of the conclusions were not well-supported by the data and/or analyses (figures and tables) they provided.

Experimental design

The research was within the scope of the journal and performed to a high technical/ethical standard, and research questions were well defined in the Introduction. As the authors stated, the molt sequences and cycles of tropical species are poorly studied, and this study fills a knowledge gap and describes interesting molt patterns that have never been previously examined. Methods were described in sufficient detail and in a replicable manner.

Validity of the findings

Figures and tables are well described in the captions and text but lack associated statistics to support the authors’ conclusions. The conclusions are linked to the original research questions but not rigorously supported by their data and analyses, and the authors need to present alternative hypotheses. Please see the comments in the “General Comments” section for details.

Additional comments

L. 102–103: It was not clear why juveniles “should” initiate primary molt at the same loci as adults if they replace the primaries in multiple series. Could they not have different nodal points?

L. 134–137: Need more explanation for the relationship between aggression and retainment of bursa.

L. 162–164: If the sample collection was not random, the data would still not be “unbiased.” Obviously, the samples were not collected randomly throughout breeding and molting seasons.

L. 212–217: The fact that only 17%–31% of adults replacing primaries in two or more waves appears to indicate that this is not their dominant molt strategy. Is it possible that only younger individuals (especially second-year birds) that have not completed the mot of distal primaries in the previous year molt in two waves?

L. 214: It is stated here that two groups of primaries were separated by one to several older primaries, and Fig. 1 shows that the number of feathers between two molt waves are less than 6 for more than 1/3 of individuals. Because of the sample size, I am not sure if the distribution shown in Fig. 1 is significantly different from the normal distribution. This seems contradictory to the later statement that the molt of P1 does not usually start until the distal wave reaches to about P6, which is the mechanism to reassure the fresh inner primaries are not replaced redundantly.

L. 221–228: I didn’t follow the logic here. If P1 is growing and P2 is new, should the direction of molt be ambiguous as P1 is the first primary? Same for P10.

L. 222: The phrase “P1 being too short to be compared for wear and fading with its adjacent inner primaries” does not make sense as P1 doesn’t have adjacent inner primaries.

L. 254–258: The overlap of breeding and primary molt appears to occur within a small window based on the supplemental raw data provided (mid July to early August). Most of birds do not molt during the most intense period of breeding season. I understand that the molt may be arrested during that time, but based on the data author presented, I am not convinced that the arrests during the breeding season “cause” the multiple waves of the primary molt.

L. 256: The authors do not show statistically that the arrested primary replacement was “strongly” associated with having an active crop gland.

L.271: Regarding the sentence starting “Two birds directly support…,” I feel the data did not support their conclusion exclusively as data on other birds showed that molt on P1 started before the distal wave reached P6 (Supplemental data). Since there are many birds that do not molt primaries in two waves, it seems that these examples instead support that the molt arrests during the intense breeding period do not “cause” the two waves of molt.

L. 282: Need to show the statistics if the authors are claiming “significantly positive.”

L. 282–289: These two flat lines are clearly the artifacts of sampling in July and September intensely. I agree that their molt is little synchronized, and it would be difficult to estimate molt duration reliably. However, the figure seems to indicate that most birds were in the latter half stages of molt by September. Does this indicate that their molt cycles designed to complete their primary molt before winter for most birds?

L. 324: Need statistics if you are stating that molt arrests are “strongly associated” with active crop glands.

L. 369–371: The “surprisingly” greater frequency of growing feathers in the outer than in the inner primaries in adults (Table 4) corresponds well with the greater frequency of growing feathers in the inner primaries in juveniles (Table 2). This made me think that some of the “adults” replacing primaries in two waves are second year birds that failed to replace the outer primaries in the hatch year.

Figure 2: It would be helpful if you could mark the individuals that were molting primaries in two or more waves.

---

## Round 0.2 · accepted · Accept

I have to apologize because it took me longer to review your revisions. However, I have now reviewed your response to the reviewers very thoroughly and feel that you have addressed their concerns quite well. I am happy to accept the manuscript at this point!